Genetic modifiers of response to thalidomide in transfusion-dependent beta-thalassemia patients: a whole-exome sequence analysis

Bawazir Waleed Mohammed 1 2
Khan Muhammad Ihtesham 3 4
http://orcid.org/0000-0002-6321-8248 Hazzazi Mohannad Saeed 1 2 5
Basabrain Ammar Abdullah 1 2
http://orcid.org/0000-0001-7841-7058 Khan Muhammad Tariq Masood 6
Siraj Sami 7
Almashjary Majed Naser 1
http://orcid.org/0000-0002-1578-3998 Radhwi Osman 1 8
Harakeh Steve 9
Yousafzai Yasar Mehmood 4 10 yasaryousafzai@gmail.com
1 Hematology Research Unit, King Fahd Medical Research Center, King Abdul Aziz University , Jeddah , Saudi Arabia
2 Department of Medical Laboratory Sciences, Faculty of Applied Medical Sciences, King Abdulaziz University, King Abdul Aziz University , Jeddah , Saudi Arabia
3 Department of Pathology, Khyber Medical College , Peshawar, Khyber Pakhtunkhwa , Pakistan
4 Institute of Pathology and Diagnostic Medicine, Khyber Medical University , Peshawar , Pakistan
5 King Fahd Medical Research Center, Animal House Unit, King Abdulaziz University , Jeddah , Saudi Arabia
6 Department of Medicine, Pak International Medical College , Peshawar , Pakistan
7 Institute of Pharmaceutical Sciences, Khyber Medical University , Peshawar , Pakistan
8 Hematology Department, Faculty of Medicine, King Abdul Aziz University , Jeddah , Saudi Arabia
9 Yousef Abdul Latif Jameel Scientific Chair of Prophetic Medicine Application, Faculty of Medicine, King Abdul Aziz University , Jeddah , Saudi Arabia
10 Department of Pathology, Rehman Medical Institute , Peshawar , Pakistan
Ozdag Hilal
Electronic publication date: 2025 Oct 7
Publication date: 2025
Volume: 13
Electronic Location ID: e20038
Received 2025 Jun 4; Accepted 2025 Aug 14
Copyright: © 2025 Bawazir et al.
Copyright year: 2025
Copyright holder: Bawazir et al.
License: This is an open access article distributed under the terms of the Creative Commons Attribution License, which permits unrestricted use, distribution, reproduction and adaptation in any medium and for any purpose provided that it is properly attributed. For attribution, the original author(s), title, publication source (PeerJ) and either DOI or URL of the article must be cited.
License URL: https://creativecommons.org/licenses/by/4.0/

Keywords: CHI3L1, LGR6, NPNT, rs880633, rs35132891, rs10425763, rs7518979, rs3207618, Transfusion dependent thalassemia, Thalidomide

Funding: Deanship of Scientific Research (DSR) at King Abdulaziz University (KAU) G: 680-142-1443 The funds were provided by the Deanship of Scientific Research (DSR) at King Abdulaziz University (KAU) under grant number (G: 680-142-1443). The funders had no role in study design, data collection and analysis, decision to publish, or preparation of the manuscript.

==============================
Background

Thalidomide induces fetal hemoglobin and renders most thalassemia patients transfusion-independent. Some patients, however, do not respond. Underlying genetic variations responsible for variable responses to thalidomide are unexplored.

Aims and objectives

To discover genetic variations that influence response to thalidomide in transfusion-dependent beta-thalassemia patients.

Methods

Twenty beta-thalassemia patients (14 excellent responders and six non-responders) who had received thalidomide were included in the study by a non-probability purposive sampling technique. Patients who showed a rise of >2 mg/dl in hemoglobin level and/or whose hemoglobin levels reached 9 gm/dl without blood transfusions were designated as excellent responders. Patients whose hemoglobin levels did not show an increment rise of >2 and/or whose hemoglobin levels did not rise above 5.9 gm/dl and needed blood transfusions to maintain optimal hemoglobin levels were designated as non-responders. DNA was extracted, and whole-exome sequencing was performed on an Illumina HiSeq System. Aligning and variant calling were done by the Sentieon software. Annotation was done by Annovar.

Results

The age of study participants ranged from 1–12 years, with a mean of 5.45 ± 3.81 years. There were 17 (85%) males and three (15%) females. A total of 222,180 germline variants were identified across 20 subjects, from which 24 candidate variants across 24 genes were identified. The three most common polymorphisms in the excellent responder group were found in the exon region of CHI3L1 (rs880633), NPNT (rs35132891), and ZNF 208 (rs10425763), which were found in 92%, 85%, and 71% cases, respectively. The commonest polymorphisms in the non-responder group were found in the PM20D1 gene (rs7518979), LGR6 (rs75658797), MYH15 (rs4299484), and RESF1 (rs3207618), each of which was found in 66.6% cases.

Conclusion

This study shows a significant association of single-nucleotide polymorphisms rs880633, rs35132891, and rs10425763 with excellent response status, while rs7518979, rs75658797, rs4299484, and rs3207618 are associated with non-response status.

Introduction

Transfusion-dependent beta-thalassemia (TDT) patients require regular blood transfusions to maintain optimal hemoglobin levels (Musallam et al., 2021). The worldwide carrier rate of TDT is 1.5%. The highest carrier rate in the Southeast Asian region is found in Malaysia (12%), followed by India (8%) and then Pakistan (5–7%) (Khaliq, 2022; Shah et al., 2019). Stem cell transplant is the only cure, but a lack of compatible donors and its high cost are the problems (Cappellini et al., 2018; Musallam et al., 2021). Splenectomy, aimed at improving the red cell count and hence hemoglobin levels, is associated with sepsis and thrombosis, which makes it a less favorable option (Cappellini et al., 2018).

Current research in low-middle income countries (LMICs) is focused on discovering drugs that increase hemoglobin-F levels, rendering TDT patients transfusion-independent (Cappellini et al., 2014; Musallam et al., 2021; Taher et al., 2014). Thalidomide is an immunomodulatory drug that induces fetal hemoglobin (Musallam et al., 2021). However, the exact mechanism is yet to be discovered (Cappellini et al., 2018; Hagh et al., 2011). So far, researchers have only been able to report altered expression of genes in patients taking thalidomide (Hagh et al., 2011). Chen et al. (2017) has proposed that thalidomide increases levels of reactive oxygen species mediated by p38 MAPK signaling and histone H4 acetylation. However, this proposed mechanism fails to explain the raised hemoglobin-F by thalidomide. Yang et al. (2020) reported the association of certain single nucleotide polymorphisms (SNPs) in HBG2 and BCL11A gene with high fetal hemoglobin levels in thalassemia patients. Hagh et al. (2011) observed decreased expression of NF-kB in thalassemia patients taking thalidomide and hypothesized that thalidomide induces fetal hemoglobin by suppressing the expression of NF-kB. Liang et al. (2021) observed that the knockout of the GATAD2A gene caused increased levels of fetal hemoglobin and proposed that thalidomide acts by suppressing this gene. Despite extensive research, the exact mechanism of fetal hemoglobin induction by thalidomide remains unexplained.

Up to 30% of TDT patients do not respond to thalidomide (Bhattacharjee et al., 2023; Garg et al., 2023). Several possible mechanisms can exist. For instance, thalidomide pharmacokinetics and pharmacodynamics, the nature of the HBB mutation itself, or mechanisms related to gamma-chain repression and induction. Therefore, an unbiased discovery-based approach is more appropriate compared to a targeted approach. In this regard, whole-exome sequencing analysis would be a smart modality to precisely and comprehensively study all the genetic variations in protein-coding genes throughout the genome (Suhaimi et al., 2022).

In this first-of-its-kind study, we enrolled TDT patients and stratified them into excellent and non-responders to thalidomide. Genetic variations between the two groups were observed and documented as genetic modifiers or prognostic factors for response to thalidomide. They should be sequenced in all TDT patients before starting therapy to predetermine their response status.

Materials and Methods

This cross-sectional study was conducted as per the ethical principles outlined in the Declaration of Helsinki 1975 and after obtaining ethical approval from the Khyber Medical University (KMU) ethical committee (letter No: KMU/IPDM/IEC/2021/09). The study was conducted at the Institute of Pathology and Diagnostic Medicine (IPDM), Khyber Medical University (KMU), Peshawar from July 2021 to July 2023. Twenty TDT patients were recruited from two different centers i.e., “Blood Disease Clinic” and a “Thalassemia Clinic” using a non-probability purposive sampling technique. TDT patients of both genders and all ages who were given thalidomide for not less than 6 months’ duration were included while those with concomitant hematological disorders like leukemia, and those who had been transfused within 2 months from the date of determining their response status were excluded from the study. Compound heterozygous cases showing bands for hemoglobin-E, hemoglobin-C, and hemoglobin-S on electrophoresis were also excluded. The cases were followed up every month for up to 6 months to determine the response status. The laboratory investigations performed included complete blood counts, liver function tests, and renal function tests.

A written informed consent was obtained from the parents of the participants aged 1–6 years. Assent was obtained in cases where participants were aged 7–17 years of age. The research work was explained to the participants and their parents, ensuring them of the confidentiality of their data. They were informed that they could withdraw at any time from the research. They were also informed that the data will be disseminated for the purpose of adding knowledge to the existing literature and it will help in the treatment of thalassemia patients all over the world.

Sample collection

Thalidomide was given to patients at a dose of 1–4 mg/kg body weight once daily as part of their treatment protocol in the thalassemia clinic by the consultant hematologist. The dose was based on the experience of the consultant hematologist where it was observed that most of the patients responded to the minimum dose of 1 mg/kg body weight. Patients who do not respond to the dose of 4 mg/kg body weight show no response to further increases in the dose. The drug was started at a minimum dose of 1 mg/kg body weight. Patients who did not respond at the lower doses were gradually subjected to higher doses every month until the maximum dose of 4 mg/kg weight was achieved. Response status was assessed after a 6-month duration (Bhattacharjee et al., 2023). Patients who required blood transfusions during the follow-up period were transfused the red cell concentrates as needed. It was ensured that the cases where the last transfusion happens within a 1-month time frame of the time of determining the response status are excluded. Patients who showed an increment rise of >2 mg/dl in hemoglobin level and/or whose hemoglobin levels reached 9 gm/dl without blood transfusions were designated as excellent responders. Patients whose hemoglobin levels did not show an increment rise of >2 and/ or whose hemoglobin levels did not rise above 5.9 gm/dl and needed blood transfusions to maintain optimal hemoglobin levels despite using thalidomide were designated as non-responders. A minimum sample size of 16 was calculated by the Open Epi sample size calculator using the worldwide incidence of transfusion-dependent beta-thalassemia of 1 per 100,000 individuals (Ali et al., 2021), absolute precision of 5% and design effect of 1.0. We, however, enrolled 20 TDT patients, which consisted of six non-responders and 14 excellent responders (Table 1). The non-responders and excellent responders were standardized with respect to age and gender.

Table 1 Characteristics of study participants.

Responder status	Age	Gender	Address	Ethnicity	Pathogenic mutations in TDT*	Transfusion frequency (pre-thalidomide)	Duration of use of thalidomide	Dose of thalidomide	Hb before starting thalidomide	Hb at the 6th month of thalidomide therapy	Increment increase in Hb levels	
NR1	7 years	M	Dera Ghazi Khan	Wihar	Homo IVS1-5	2 weeks	1 year	50 mg	5.1	6.9	1.8	
NR2	12 years	M	Dera Ghazi Khan	Baloch chandia	Homo IVS1-5	2 weeks	2 year	60 mg	6.5	7.2	0.7	
NR3	2 years	M	Dera Ghazi Khan	Laghari Baloch	Homo IVS1-5	4 weeks	1 year	40 mg	3.2	5.2	2	
NR4	10 years	M	Data Khel	–	Homo Fr 8-9	7 weeks	1 year	50 mg	6.4	7.1	0.7	
NR5	2 years	M	Shamsatoo	Afridi	Homo Fr 8-9	3 weeks	1 year	50 mg	5	5.5	0.5	
NR6	4 years	F	Dera Ghazi Khan	Baloch chandia	Homo IVS1-5	2 weeks	1 year	50 mg	5.8	6	0.2	
ER1	3 years	M	Sialkot	Jutt	Homo IVS1-5	4 weeks	8 months	30 mg	6	10	4	
ER2	2 years	F	Swabi	Yousafzai	Homo Fr 8-9	6 weeks	1 year	20 mg	5.9	9	3.1	
ER3	5 years	M	Jampur	–	Homo IVS1-5	4 weeks	1 year	30 mg	4.5	9.3	4.8	
ER4	11 months	M	Lahore	Shinwari	Homo Fr 8-9	3 weeks	1 year	10 mg	6.5	9.6	3.1	
ER5	3.5 years	M	Gujranwala	Sheikh	Homo Fr 8-9	4 weeks	7 months	30 mg	7	10.1	3.1	
ER6	5 years	M	Nowshehra	–	Homo Fr 8-9	4 weeks	1 year	30 mg	7	9.2	2.2	
ER7	5 years	M	Tarbella	Mughal	Homo Fr 8-9	4 weeks	1 year	50 mg	6.1	9	2.9	
ER8	7 years	M	Attock	Awan	Homo IVS1-5	6 weeks	5 months	30 mg	6	11.9	5.9	
ER9	9 years	M	Khanewal	–	Homo Fr 8-9	6 weeks	1 year	50 mg	6.2	9.7	3.5	
ER10	1 years	M	Swabi	Dhobian	Homo Fr 8-9	5 weeks	1 year	10 mg	6.5	10	3.5	
ER11	1 years 4 months	M	Peshawar	Tarkaan	Homo Fr 8-9	4 weeks	10 months	10 mg	6.2	9.3	3.1	
ER12	6.5 years	F	Kashmir	Qureshi	Homo Fr 8-9	3 weeks	1 year	30 mg	6	10.2	4.2	
ER13	12 years	M	Kashmir	–	Homo Fr 8-9	6 weeks	6 months	40 mg	4.4	10	5.6	
ER14	12 years	M	Karachi	Afridi	Homo Fr 8-9	4 weeks	1 year	30 mg	6.1	11.2	5.1	
Notes:

ER, Excellent responder; NR, non-responder; M, male; F, female; Homo IVS1-5, homozygous for IVS1-5 mutation; Homo Fr 8-9, homozygous for Frameshift 8-9 mutation; Hb, hemoglobin (in gm/dl); Ethnicity data of NR4, ER3, ER9 & ER12 was not available.

* Determined by PCR as a baseline before starting thalidomide therapy.

Two milliliters of venous blood sample were drawn from the participants in a purple top tube. DNA was extracted from the samples by the salting-out method (MWer, Dykes & Polesky, 1988). The extracted DNA samples were checked for the quantity of DNA by NanoDrop and gel electrophoresis. Samples with a DNA concentration of >50 ng/microliter and showing clear bands on gel electrophoresis were subjected to whole-exome sequencing.

Whole-exome sequencing and data pre-processing

The detail of whole-exome sequencing is explained in File S1. The quality control of exome data along with depth and coverage details is shown in File S2. The raw data was optimized by Cutadapt software (version 1.9.1). Clean data was aligned with reference human genome hg19/GRCh37 using Sentieon, and then sequenced and regenerated into BAM files. Variant calling was performed by Sentieon software (GATK haplotype caller) and annotation by Annovar (Wang, Li & Hakonarson, 2010). Differentially mutated genes were searched throughout the exome by search strategy (details given in File S3). The mutations were visually confirmed by uploading BAM files into Integrated Genome Viewer (IGV) software and documenting the mutations (shown in File S4).

Variant identification

For all samples, the data quality was robust, with over 85% of effective data, Q20 scores exceeding 90%, Q30 scores surpassing 80%, and a GC content of around 50%. Among the qualified samples, the mapping rate reached 95%. The mean target coverage stood at 90%, with a mean target depth of no less than 60x. In total, 222,180 germline variants were identified across 20 subjects, comprising 188,660 SNVs and 23,503 INDELs. To pinpoint functional mutations, stringent criteria were applied. Initially, splice site, nonsense, and frameshift mutations were isolated, followed by the inclusion of functional missense variants through bioinformatic annotation tools. From the refined variant pool, 24 candidate variants across 24 genes were identified (refer to File S5). Among these variants, 15 exhibited a nonsynonymous mutational burden of 0.99 in our 24 single nucleotide variants (SNVs).

Functional analysis of candidate variant-associated genes

To determine the underlying biological process and pathways affected by the significant genes identified by variants associated with TDT, functional enrichment analyses were performed. Biological process enrichment analysis from variant-associated genes was performed by using DAVID (Shannon et al., 2003). In addition, we used the Kyoto Encyclopedia of Genes and Genomes (KEGG) pathways database to perform pathway enrichment analysis of consensus mutations in TDT patients. The most statistically enriched GO terms were visualized using the ggplot2 visualization package (Wickham, 2011).

Statistical analysis

The allele frequencies were estimated by the gene counting method. The difference in mutations between genes in the two groups was assessed for significance using Fisher’s test using SPSS (ver. 22; IBM Corp., Armonk, NY, USA). A p-value of <0.05 was taken as significant. The proteins encoded by the genes were studied in detail using the UniProt database. Quantitative variables were analyzed by mean and standard deviation. Frequency and percentages were used for qualitative variables.

Results

A total of 20 TDT patients were included in the study. The mean age of study participants was 5.45 ± 3.81 years (range: 1–12 years). There were 17 (85%) males and 3 (15%) females, male to the male-to-female ratio being 5.6:1. Characteristics of study participants are given in Table 1.

In total, 222,180 germline variants were identified across 20 subjects, comprising 188,660 SNVs and 23,503 INDELs. To pinpoint functional mutations, stringent criteria were applied. Initially, splice site, nonsense, and frameshift mutations were isolated, followed by the inclusion of functional missense variants through bioinformatic annotation tools. From the refined variant pool, 24 candidate variants across 24 genes were identified (refer to File S5). Among these variants, 15 exhibited a nonsynonymous mutational burden of 0.99 in our 24 SNVs (Table 2).

Table 2 Genes mutated differentially between excellent responders and non-responders as determined by whole exome sequencing analysis.

S.
No	Mutated Genes	Chromosome number	Region of gene	Reference nucleotide	Alternate nucleotide	Position
(with respect to GRCh 37/hg19)	HGVS Accession number with variant detail	db. SNP ID	Prevalence of mutation		
Excellent responders
(n = 14)	Non-responders
(n = 6)	p-value
(Fischer’s exact test)	
1	NPNT	4	Exon	G	C	106,859,549	NC_000004.11:g.106859549G>C	rs35132891	12/14 (85%)	NA	0.001	
2	BMP3	4	Exon	G	A	81,967,150	NC_000004.11:g.81967150G>A	rs3733549	9/14 (64%)	NA	0.014	
3	CHD4	12	Exon	C	A	6,711,147	NC_000012.11:g.6711147C>A	rs1639122	9/14 (64%)	1/6 (16.6%)	0.141	
4	MAPK4	18	Exon	G	A	48,190,440	NC_000018.9:g.48190440G>A	rs3752087	9/14 (64%)	NA	0.014	
5	TAF1C	16	Exon	A	T	84,213,434	NC_000016.9:g.84213434A>T	rs2230129	11/14 (78%)	1/6 (16.6%)	0.018	
6	INPP5D	2	Exon	A	G	233,990,594	NC_000002.11:g.233990594A>G	rs61752227	10/14 (71%)	NA	0.011	
7	ACP1	2	Exon	C	T	272,203	NC_000002.12:g.272203C>T	rs11553746	11/14 (78%)	1/6 (16.6%)	0.018	
8	LGR6	1	Exon	G	A	202,287,754	NC_000001.10:g.202287754G>A	rs75658797	NA	4/6 (66.6%)	0.003	
9	HHAT	1	Exon	G	A	210,577,884	NC_000001.10:g.210577884G>A	rs2294851	NA	3/6 (50%)	0.018	
10	CHI3L1	1	Exon	T	C	203,152,801	NC_000001.10:g.203152801T>C	rs880633	13/14 (92%)	NA	0.000	
11	KHNYN	14	Exon	A	C	24,901,249	NC_000014.8:g.24901249A>C	rs3742520	5/14 (35%) hetro.
1/14 (7%) homo.	6/6 (100%) homo.	0.042	
12	ZNF208	19	Exon	T	C	22,155,918	NC_000019.9:g.225918T>C	rs10425763	10/14 (71%)	NA	0.011	
13	MYH15	3	Exon	C	T	108,189,627	NC_000003.11:g.108189627C>T	rs4299484	NA	4/6 (66.6%)	0.003	
14	RESF1	12	Exon	G	A	32,134,926	NC_000012.11:g.32134926G>A	rs3207618	NA	4/6 (66.6%)	0.003	
15	BIRC2	11	Exon	C	T	102,248,377	NC_000011.9:g.102248377C>T	rs34510872	1/14(7%)	4/6 (66.6%)	0.014	
16	PM20D1	1	Exon	A	G	205,812,912	NC_000001.10:g.205812912A>G	rs7518979	NA	5/6 (83.3%)	0.000	
OFF-TARGET GENES	
17	CCDC18	1	UTR5	C	T	93,646,418	NC_000001.10:g.93646418C>T	rs10874763	NA	4/6 (66.6%)	0.003	
18	CXCR4; THSD7B	2	Intergenic	G	T	137,087,148	NC_000002.11:g.137087148G>T	rs72847680	NA	4/6 (66.6%)	0.003	
19	SNHG14	15	NcRNA_exonic	T	C	25,219,512	NC_000015.9:g.25219512T>C	rs705	9/14 (64%)	NA	0.014	
20	ACSM2B; ACSM1	16	Intergenic	C	T	20,605,255	NC_000016.9:g.20605255C>T	rs201994972	10/14 (71%)	NA	0.042	
Intergenic	CT	C	20,599,458	NA	rs149883706	11/14 (78%)	NA	0.014	
Note:

Homo, Homozygous; Hetro, heterozygous; UTR5, 5′ untranslated region; Nc, non-coding.

The three most common SNPs in excellent responder group were found in exon region of CHI3L1 (rs880633), NPNT (rs35132891) and ZNF 208 (rs10425763) in 13/14 (92%) (p-value = 0.000), 12/14 (85%) (p-value = 0.001) and 10/14 (71%) (p-value = 0.011) excellent responders respectively. The commonest SNPs in non-responder group were found in exon region of PM20D1 gene (rs7518979) which was mutated in 5/6 (83.3%) (p-value = 0.000) non-responders, followed by LGR6 (rs75658797), MYH15 (rs4299484), and RESF1 (rs3207618) each of which were mutated in 4/6 (66.6%) (p-value = 0.003) non-responders (Table 2).

The functional enrichment analysis indicated that the genes of identified candidate variants in TDT were mainly enriched in biological processes related to thalassemia response such as IL-6, NF-kappa-B signaling, etc. (Fig. 1A). Identified rare variant-associated genes were subjected to pathway enrichment analysis based on gene ontology, KEGG (Fig. 1B). The majority of the pathways represented are associated with TDT such as Apoptosis, IL-17 signaling pathways, and B-cell receptor signaling pathways, lipid, folate, and cholesterol metabolism which are highly associated with TDT.

Figure 1 Representative results for enrichment analyses of candidate variants associated with TAT.

(A) Significantly enriched GO terms from biological processes. (B) Significantly enriched KEGG pathways.

Discussion

To our knowledge, this is the first study that explored genetic variations associated with varying responses to thalidomide in TDT patients using whole-exome sequencing. In our study, genetic variations in NPNT, BMP3 & MAPK4 genes were seen in excellent responders. Currently, there is no established role of these genes in ineffective erythropoiesis in thalassemia. Additionally, in our study, genes known to participate in pro-apoptotic pathways and endoplasmic reticulum (ER) stress (TAF1C, INPP5D, ACP1) were mutated in the excellent responders, while the anti-apoptotic gene (BIRC2) was mutated in the non-responder group. This data hints towards genes that should be explored to understand the role of these genes as prognostic markers in the response of thalassemia patients to thalidomide.

The genetic variations in the genes highlighted in this study provide data for further transcriptome studies. This will help validate the association of these genes with response status to thalidomide in thalassemia patients and to explore the role of these genes in erythropoiesis.

The NPNT gene encodes an extracellular matrix protein, nephronectin, that binds to integrin α8β1. It regulates organogenesis and cell differentiation (Huang & Lee, 2005; Sopel et al., 2022). Recently, NPNT gene expression has been known to be regulated by the TNF-NFKB pathway and TGFβ-SMAD pathway (Fang et al., 2010; Sopel et al., 2022; Tsukasaki et al., 2012). However, the role of NPNT in erythropoiesis and hemoglobin synthesis is unknown (Magnussen et al., 2021). Clinical significance of SNP rs35132891 in the NPNT gene is not reported in Clinvar.

BMP3, the most abundant BMP in bone marrow, encodes a protein that binds to activin receptor type IIB, to activate SMAD2 which then regulates apoptosis and cell cycle (Wen et al., 2019). The clinical significance of SNP rs3733549 reported in our study is not reported in ClinVar. To our knowledge, there is no data available regarding the role of the BMP3 gene in erythropoiesis and thalassemia.

Disruption of the CHD-4 gene (components of the Nucleosome Remodeling and Deacetylase (NuRD) complex) is known to increase the fetal hemoglobin levels (Hossain et al., 2018). The SNP rs1639122 is reported as benign in ClinVar but in our study, it is associated with excellent response status. Only one study has shown a significant association of this SNP with bone mineral density (Liu et al., 2020).

The TAF1C gene encodes a protein that is involved in the synthesis of ribosomal RNA (rRNA), thus playing a role in processes like protein synthesis, cell differentiation, and apoptosis (Abdulwahab et al., 2019; Ferreira et al., 2020; Saproo, 2023). The role of the TAF1C gene in protein synthesis still needs further research (Saproo, 2023). We suggest research on the role of TAF1C in hemoglobin synthesis in thalassemia patients.

INPP5D encodes a 5′-phosphatase restricted to hematopoietic tissue. The role of INPP5D in erythropoiesis is unclear and needs further research (Mason et al., 2000). In our study, the SNP rs61752227 in the INPP5D gene was significantly associated with excellent response. However, the Clinical significance of SNP rs61752227 is not reported in ClinVar.

The SNP rs11553746 in the ACP1 gene as reported in our study, is reported as a missense variant in ClinVar. Recently, the role of the ACP1 gene in apoptosis and endoplasmic reticulum stress in HEPG2 cells has been reported (Bourebaba et al., 2020). Based on the finding that the ACP1 gene is mutated in excellent responders in our study, we suggest further studies to determine any possible role of the gene in ER stress-induced apoptosis in erythroid cells in thalassemia.

LGR6 receptors bind ligand R-spondins to activate the canonical Wnt-β-catenin pathway (Park et al., 2018). It is reported to be a tumor suppressor gene (Schuebel et al., 2007). Activation of LGR6 receptors on macrophages by maresin1 promotes phagocytosis and efferocytosis by macrophages (Chiang et al., 2019). Clinical significance of the SNP rs75658797 is not reported in ClinVar. However, we report that it is significantly mutated in non-responders.

CHI3L1 is found to be localized in the ER and its depletion, as seen in CHI3L1 knockout mice, increases superoxide dismutase-1 level which then activates ER-induced apoptosis in cells (Yu et al., 2023). We suggest further studies to evaluate any possible role of CHI3L1 in ER stress-induced apoptosis in ineffective erythropoiesis in thalassemia. In our study, the SNP rs880633 was present in excellent responders. However, its significance is not reported in ClinVar. However, the SNP rs880633 is reported to be associated with an increased risk of cystic fibrosis, chronic obstructive pulmonary disease, and hepatocellular carcinoma, while a decreased risk of developing deformities in patients with scoliosis (Hector et al., 2011).

ZNF208 is a member of the zinc finger gene family (Eichler et al., 1998). Generally, ZNF proteins bind to specific areas of DNA or RNA and regulate gene expression. We report the SNP rs10425763 in the ZNF208 gene to be associated with excellent response to thalidomide. However, the significance of the SNP is not reported in ClinVar. However, certain other variants of ZNF208 were reported to be associated with stroke and certain cancers (Alanazi & Iqbal, 2022; Wang et al., 2016; Yu et al., 2017).

The role of the MYH-15 gene in hematopoiesis is unknown and needs further workup. We report the SNP rs4299484 in the MYH15 gene to be associated with non-response status to thalidomide. A recent study established evidence for cross-talk between TGF-beta and MYH15 genes by showing that lung cancer cells, upon stimulation by TGF-beta showed increased expression of the MYH15 gene (Gladilin et al., 2019). This shows there is a link between the TGF-beta pathway and MYH15 gene expression.

RESF1 is an uncharacterized gene in humans. The gene is predicted to play a role in histone modification and gene silencing in association with the SETDB1 gene in mouse embryonic stem cells (Fukuda et al., 2018). The SNP rs3207618 is not reported in ClinVar. However, only one study reports this SNP to be associated with thalidomide-induced peripheral neuropathy in multiple myeloma patients (García-Sanz et al., 2017).

BIRC2 inhibits the non-canonical NF-KB pathway, a pathway that is necessary for the survival, proliferation, and differentiation of macrophages and other immune cells (Zadoroznyj & Dubrez, 2022). Loss of the BIRC2 gene is reported to lead to apoptotic cell death (Hrdinka & Yabal, 2019). Cellular stresses like hypoxia, ER stress, and DNA damage increase the expression of BIRC2 to promote apoptosis. However, research in erythroid cell lines in this regard has not been done.

The PM20D1 gene is involved in the regulation of body mass, temperature, insulin resistance, blood pressure, bone health, and angiogenesis. The precise functions of the body are however not fully known and need further research (Benson et al., 2019; Kim et al., 2020; Long et al., 2018). The gene has been reported as a quantitative trait locus associated with Alzheimer’s disease (Sanchez-Mut et al., 2018). There is no research regarding the function of the PM20D1 gene in hemoglobinopathies. We suggest studies to define the role of the P20D1 gene in thalassemia patients.

The major limitation of this study was the limited number of patients. This is attributed to the high cost of exome sequencing analysis. Baseline fetal hemoglobin levels were not monitored and compared in the two groups studied. A critical limitation of the study is the functional assay to confirm how these variants causally affect response to thalidomide. Further, inclusion of the pediatric population limits the generalization of findings to this subgroup only and not to the adult population.

We recommend larger studies to validate the findings of the study. Furthermore, functional studies like gene expression analysis or cellular assays should be done to establish the causal association of these variants with the response status to thalidomide.

Conclusions

This is the first study that shows that SNP rs880633, rs35132891, and rs10425763 are significantly associated with excellent response status to thalidomide in thalassemia patients, while rs7518979, rs75658797, rs4299484, and rs3207618 are associated with non-response status. We hypothesize for the first time that these genetic variants are the genetic modifiers of response to thalidomide in thalassemia patients and should be sequenced in all transfusion-dependent β-thalassemia patients before starting thalidomide to predetermine their response status. Only excellent responders will be continued on low-dose thalidomide therapy. However, gene expression analysis studies are recommended to establish these SNPs as the prognostic factors of response to thalidomide in thalassemia patients.

Supplemental Information

Supplemental Information 1 Whole exome sequencing method.

Supplemental Information 2 Quality control files.

Supplemental Information 3 Search strategy.

Supplemental Information 4 visual depiction of mutations by Integrated Genome Viewer (IGV).

Supplemental Information 5 annotated mutations.

Supplemental Information 6 Full names of the genes.

This research work is submitted in partial fulfillment of the requirements for the Ph.D. of Muhammad Ihtesham Khan. Additionally, grammatical errors from the manuscript were removed by using Grammarly online app by uploading the manuscript file to the software and downloading the corrected version of the manuscript.

Additional Information and Declarations

Competing Interests

The authors declare that they have no competing interests.

Author Contributions

Waleed Mohammed Bawazir analyzed the data, authored or reviewed drafts of the article, and approved the final draft.

Muhammad Ihtesham Khan performed the experiments, prepared figures and/or tables, authored or reviewed drafts of the article, and approved the final draft.

Mohannad Saeed Hazzazi analyzed the data, authored or reviewed drafts of the article, and approved the final draft.

Ammar Abdullah Basabrain analyzed the data, authored or reviewed drafts of the article, and approved the final draft.

Muhammad Tariq Masood Khan performed the experiments, authored or reviewed drafts of the article, and approved the final draft.

Sami Siraj conceived and designed the experiments, authored or reviewed drafts of the article, and approved the final draft.

Majed Naser Almashjary analyzed the data, prepared figures and/or tables, and approved the final draft.

Osman Radhwi analyzed the data, prepared figures and/or tables, and approved the final draft.

Steve Harakeh analyzed the data, prepared figures and/or tables, authored or reviewed drafts of the article, and approved the final draft.

Yasar Mehmood Yousafzai conceived and designed the experiments, authored or reviewed drafts of the article, and approved the final draft.

Human Ethics

The following information was supplied relating to ethical approvals (i.e., approving body and any reference numbers):

This study was conducted after obtaining Ethical approval from the Khyber Medical University (KMU) ethical committee (letter No: KMU/IPDM/IEC/2021/09.)

Data Availability

The following information was supplied regarding data availability:

The data is available at NCBI/Clinvar: SCV006081246, SCV006081247, SCV006081248, SCV006081249, SCV006081250, SCV006081251, SCV006081252, SCV006081253, SCV006081254, SCV006081255, SCV006081256, SCV006081257, SCV006081258, SCV006081259, SCV006081260, SCV006081261, SCV006081262, SCV006081263, SCV006081264, SCV006081265.

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
