# Peer review of "Genetic modifiers of response to thalidomide in transfusion-dependent beta-thalassemia patients: a whole-exome sequence analysis"

_PeerJ, doi:10.7717/peerj.20038_

## Round 0.1 · original submission · Minor Revisions

**Language Note:** The review process has identified that the English language must be improved. PeerJ can provide language editing services - please contact us at [email protected] for pricing (be sure to provide your manuscript number and title). Alternatively, you should make your own arrangements to improve the language quality and provide details in your response letter. – PeerJ Staff

·

Basic reporting

The manuscript is so interesting with a very original topic, design and findings

- The English needs to be reviewed and improved with forming more compound sentences mainly in the introduction part.

Some Clarifications need to be mentioned for better understanding for the reader:
- It was mentioned that the response to thalidomide was assessed after 6 months, I think it is reasonable to state in the methods section how the patients were followed up during the 6 months period, what was the frequency of the labs taken and what parameters were followed up.

- on what basis was the dosage of the thalidomide chosen? Was it only physician decision? it was mentioned that the dose ranged between 1-4 mg/Kg and this a bit wide range with wide differences in the total dose of thalidomide from one patient to the other. Why the dose was not a fixed one for all patients (for example 2 mg/Kg for all or 3 mg/Kg for all)?

- Was blood transfusion allowed at any point of the study duration? if yes, what was the limit of transfusion to exclude the patient from the study?

Experimental design

- The experimental design is suitable and well chosen. However, some clarifications should be mentioned in the methods section. The comments and clarifications are mentioned in the previous section of my review.

Validity of the findings

- The findings are very interesting with a high scientific value that opens the door for further investigations and treatment modifications based on the genetic sequencing

Additional comments

- It is advised to add a table showing the baseline parameters for the participants including the level of Hb at baseline, frequency of transfusion, Hb trend over the months or every 3 months

- It is advised to add to the discussion if increasing the dose of thalidomide might overcome the resistant variants in non-responding patients

·

Basic reporting

The manuscript titled “Genetic Modifiers of Response to Thalidomide in Transfusion-Dependent Beta-Thalassemia Patients: A Whole Exome Sequencing Analysis” refers to a new research into the genetic modifiers affecting treatment responses with Thalidomide in transfusion-dependent beta-thalassemia (TDT) patients.
Clear scientific language is used to present the authors' findings. Minor changes could improve the manuscript's clarity and flow.

The introduction is well-written and emphasises the therapeutic need of understanding treatment response heterogeneity. It effectively highlights the gap about individual thalidomide response and effectively suggests whole exome sequencing (WES) as an exploratory approach.
The study aims to determine genetic variations linked to TDT patients' response or non-respone to thalidomide, which is the research question. This research is important for both medicine and science, especially in resource-limited countries where cost-effective medications are required.

Experimental design

The study’s design, although constrained by a modest sample size of twenty participants, is appropriate for an initial exploratory genomic investigation, particularly given the high costs associated with WES. Clear ethical approval, informed consent, and inclusion/exclusion criteria address ethics. The methodology details sequencing platforms (Illumina HiSeq), bioinformatics pipelines (Sentieon, Annovar), and variant prioritisation criteria strengthen data visibility.

Validity of the findings

Key findings highlight statistically significant associations between the "excellent responder" phenotype and specific single nucleotide polymorphisms (SNPs), including rs880633 in CHI3L1, rs35132891 in NPNT, and rs10425763 in ZNF208. On the other hand, different variants such as rs7518979 in PM20D1 and rs75658797 in LGR6 show a higher prevalence among non-responders. Fisher's statistical test is appropriate for statistical analysis, and the p-values support the findings. As noted by the authors, a small sample size limits statistical power and generalisation of results.From a data sharing perspective, the authors comply with open data policies by providing comprehensive raw metrics, variant lists, and supplementary materials, thereby facilitating reproducibility and further research.
The discussion interprets these genetic associations within biological contexts, considering potential roles in pathways such as IL-6, NF-κB, and TGF-β signaling, which are relevant to ineffective erythropoiesis.
Gene Ontology (GO) and KEGG pathway analysis place results in biological relevance.
The authors suggest additional research directions and recognise that many discovered SNPs in ClinVar are unannotated or of questionable significance, emphasising the further need for validation research.
A major limitation is the absence of baseline fetal haemoglobin (HbF) data, which limits our knowledge of the mechanisms behind the observed responses. Baseline HbF may be a key treatment response biomarker , and this absence should be discussed. Additionally, clarify the definition of “excellent responder” in the abstract and methods sections.

Additional comments

In conclusion, minor modifications needed to improve language clarity, explain study limitations—baseline HbF data—and explain findings interpretation. These small changes would place the manuscript in a strong position for publication, significantly contributing to the field of pharmacogenomics in hemoglobinopathies.

---

## Round 0.2 · accepted · Accept

With all the reviewers' concerns now addressed, the manuscript is ready for publication.

·

Basic reporting

The authors consulted language specialists to write a clear manuscript. The introduction emphasises the clinical and scientific importance of thalidomide response genetic modifiers, especially in areas with limited resources. Recent and relevant references, and interesting and high-quality figures and tables. The manuscript is well-organised and written for the journal.

Experimental design

The study design is appropriate for whole-exome sequencing pilot research. The limited sample size (n=20) has been explained by WES's high cost. Excellent and non-responders are selected using clear criteria.

Validity of the findings

Limitations like sample size and lack of baseline fetal haemoglobin values are disclosed in the study.